# The Coordination of Aquaculture Development with Environment and Resources: Based on Measurement of Provincial Eco-Efficiency in China

**DOI:** 10.3390/ijerph19138010

**Published:** 2022-06-30

**Authors:** Wei Yan, Changbiao Zhong

**Affiliations:** Business School, Ningbo University, 818 Fenghua Road, Jiangbei District, Ningbo 315211, China; zhongchangbiao@nbu.edu.cn

**Keywords:** China’s aquaculture, eco-efficiency, undesirable outputs, the nonpoint source pollution, the super-efficiency SBM model

## Abstract

To resolve the environmental problems of China’s aquaculture industry, we must examine the current situation and comprehensively consider aquaculture growth, resource conservation and environmental protection. Using the unit investigation and evaluation method to evaluate the nonpoint source pollution of each province, this paper calculates eco-efficiency to evaluate the coordination of environment and aquaculture growth based on the slacks-based measure directional distance function dealing with undesirable outputs. The results reveal that the eco-efficiency of aquaculture in China from 2003 to 2018 is 0.70 and obviously lower than the industry’s economic efficiency, indicating aquaculture development has not been coordinated with resources and the environment. Environmental pollution brings great loss to the economic efficiency of aquaculture. Specifically, eastern China, with the highest aquaculture output, shows the best degree of coordination, followed by western China. Six provinces or province-level municipalities, including Fujian, Shanghai, Beijing, Hainan and Tianjin, are growing soundly and rapidly, while central China exhibits the most obvious imbalance among the environment, resources and aquaculture development.

## 1. Introduction

Since the 1980s, the fisheries development strategy of “promoting fisheries by fish breeding” has been vigorously implemented in China, leading to the rapid development of the aquaculture industry. The scale and output of aquaculture have both been increasing. In 2018, the production of China’s aquaculture reached 49.9106 million tons, accounting for 77.3% of the total output of aquatic products and making China the only fisheries country with aquaculture output that exceeded fishing output. However, the environment of the aquaculture industry in China is in critical condition. Some aquaculture methods, including high-density culturing and unscientific feeding, result in many kinds of pollution around the waters and cause both decreases in aquaculture resources and increases in water eutrophication, impairing the ecological environment of aquaculture waters and the quality of aquatic products [1]. In recent years, aquaculture has even been regarded as the source of pollution in some places and, therefore, has been eliminated [2]. In 2019, China formulated and promulgated several opinions on accelerating the green development of the aquaculture industry, which meant that the dilemma faced by China’s aquaculture industry gradually evolved from an imbalance between resources and development to an imbalance among the environment, resources and development. Based on the data regarding the input of factor resources, the economic output and the pollution emissions of China’s aquaculture industry, this paper analyzes the eco-efficiency (EEI) of each province from 2003 to 2018, thereby assessing the coordination of aquaculture development with the environment and resources and provides important theoretical and practical guidance for facilitating the high-quality green development of the aquaculture industry.

Eco-efficiency, which incorporates the environment and resources into production efficiency measurement, is the input–output ratio based on economic development, resource consumption and pollutant emission, and it can reflect the comprehensive coordination among economic growth, resource utilization and environmental protection [3]. As early as the 1980s, foreign researchers began to pay attention to the relationship between aquaculture and the ecological environment. Gates et al. (1980) and Folke (1988) studied the reuse of water resources and energy consumption in the process of Salmon breeding [4,5]. Some domestic researchers such as Shi and Bi (2003), Wang et al. (2003) and Zhang (2004) pointed out that the impact on the ecological environment of the surrounding waters exerted by aquaculture itself should not be ignored in relation to the water pollution caused by external pollutants [6,7,8]. In recent years, the green development of aquaculture has generated increasing academic attention and the research methods have gradually shifted from qualitative discussion to normative empirical research. Rich research achievements have been gained in the evaluation of aquaculture technical efficiency, economic efficiency and eco-efficiency, mainly consisting of the nonparametric methods represented by data envelopment analysis (DEA) and the parametric methods represented by stochastic frontier analysis (SFA). In the efficiency evaluation of parametric methods, Asche et al. (2009) measured the technical efficiency and the influencing factors of Norwegian salmon breeding using SFA [9]. Sarker et al. (2016) found that the age and education level of farmers had remarkable effects on aquaculture efficiency through an analysis of the technical efficiency of 149 farmers in Bangladesh [10]. Sun et al. (2014) pointed out that freshwater aquaculture in China possessed a high overall technical efficiency and the technical efficiency was strongly influenced by the processing rates and disaster rates of freshwater products [11]. After conducting their research, Wan and Yang (2017) also believed that the overall technical efficiency of China’s freshwater aquaculture was pretty high, averaging 0.69, and it was witnessing diminishing returns compared to its scale [12]. Lv et al. (2020) analyzed the technical efficiency of large freshwater farmers from 2011 to 2017 using the stochastic frontier production function model, and the results showed that the total factor productivity of farmers grew at an annual rate of 3.29% and presented an increasing trend [13]. With regard to the efficiency evaluation of nonparametric methods, Martinez-Cordero and Leung (2004) took sewage discharge containing nitrogen and phosphorus as the undesirable output variable to measure the aquaculture efficiency of prawns in Mexico [14]. Artia and Leung (2014) measured the technical efficiency of aquaculture on Hawaiian farms and found that the eco-efficiency of aquaculture was lower than the technical efficiency without considering undesirable outputs [15]. Ji and Zeng (2016) analyzed the green technical efficiency of mariculture in China by using global DEA and found that the green technical efficiency has gradually increased since 2008, with Hainan Province showing the highest efficiency [16]. Qin et al. (2018) took a nonradial and nonangular slack-based measure (SBM) model to analyze the eco-efficiency of China’s mariculture, and the average eco-efficiency from 2008 to 2016 turned out to be 0.62, showing great room for improvement [17].

According to the existing studies, researchers around the world have made a great contribution to the measurement of the technical efficiency and eco-efficiency of aquaculture; however, there are still three limitations. First, when it comes to environmental factors, researchers mainly take environmental pollution as the undesirable output but fail to properly measure aquaculture pollution. For instance, Qin et al. (2018) only took the pollution output for a part of the mariculture industry as the environmental impact variable, which could not fully reflect the pollution output of aquaculture. Ji et al. (2016) took the economic loss caused by environmental pollution as the undesirable output variable, but some variables were only estimated due to the lack of corresponding statistical data. Second, aquaculture consists of freshwater aquaculture and mariculture. The existing studies on the eco-efficiency of aquaculture in China mainly focus on mariculture rather than aquaculture overall. Third, the existing research has not further divided and ranked the full-efficiency production units (efficiency = 1) which were measured via DEA.

In this paper, we try to extend the existing studies in the following three aspects. First, the unit survey and assessment method is applied to accurately calculate the provincial pollutant emissions of aquaculture, termed the emissions of chemical oxygen demand (COD_cr_), total nitrogen (TN) and total phosphorus (TP), which are the basis for eco-efficiency evaluation. Then, we apply the SBM model approach dealing with undesirable outputs to calculate the eco-efficiency of the overall aquaculture industry (i.e., freshwater aquaculture and mariculture) accounting for environmental regulations at the provincial level in China from 2003 to 2018. Third, the super-efficiency DEA model is applied to allow the eco-efficiency of full-efficiency production units to be greater than or equal to 1, making it possible to rank full-efficiency production units.

## 2. Methods

Compared with SFA, DEA has 2 distinct advantages. First, there is no need to set specific functional form, avoiding the problems caused by improper setting of production function, and there is no need for dimensionless variables. Second, DEA can handle systems with multi-inputs and multi-outputs. Liu et al. (2010) believed that the best current way to measure eco-efficiency was constructing an input distance function with undesirable output using the SBM model [18]. Therefore, this paper measures the eco-efficiency using an optimized SBM model [19,20], thereby analyzing the coordination of the environment, resources and aquaculture in various provinces of China.

The DEA model, first proposed by Charnes et al. (1978), is mainly used to evaluate the relative efficiency between decision-making units (DMUs) with multi-inputs and multi-outputs by mathematical programming model [21]. Because the DEA model does not need to set specific function form and is not affected by the dimension of input and output variables, its evaluation results are objective and the DEA model has been widely used in efficiency evaluation. However, the traditional DEA model calculates the relative efficiency from the radial perspective, without considering the slack of input and output, which often leads to higher results and affects calculation accuracy. To solve the above problems, Tone (2001) proposed a nonradial and nonangular DEA model, namely the SBM model [22]. This model can effectively avoid the inherent problem of slack and angle selection of the traditional DEA model, improve the accuracy of results and conduct the efficiency evaluation while the undesirable output exists. In this study, we use an optimized SBM model to calculate the eco-efficiency of aquaculture.

According to Fare et al. (1994), environmental production technology can be presented via the DEA model [23]. Assuming input–output vector of k=1,…,K production units = ($x_{{}_{k}}$, $y_{k}$, $b_{k}$), then:(1)$$P(x)=\left[\begin{array}{l} (y,b):\sum_{k=1}^{K}z_{k}y_{m}^{k}\geq y_{m},m=1,\ldots ,M;\sum_{k=1}^{K}z_{k}b_{i}^{k}=b_{i}^{},i=1,\ldots ,I;\\ \sum_{k=1}^{K}z_{k}x_{n}^{k}\leq x_{n}^{},n=1,\ldots ,N;z_{k}\geq 0,k=1,\ldots ,K \end{array}\right]$$

Formula (1) is environmental production technology with constant returns to scale, expressed by density vector $z_{k}\geq 0$, indicating the respective weights of k=1,…,K production units when constructing the environmental technology structure. Furthermore, assuming:(2)$$\sum_{k=1}^{K}b_{i}^{k}>0,i=1,\ldots ,I;\sum_{i=1}^{I}b_{i}^{k}>0,k=1,\ldots ,K$$

Formula (2) represents the following 2 points, respectively: At least 1 production unit produces each type of undesirable output, and each production unit produces at least 1 type of undesirable outputs.

Under the condition of formula (1), we construct a nonradial and nonangular SBM model with undesirable outputs:(3)$$\begin{array}{l} \rho^{*}=\min\rho =\underset{}{\min}\frac{1-\left[\frac{1}{N}\sum_{n=1}^{N}\frac{s_{n}^{x}}{x_{n}^{k^{\prime }}}\right]}{1+\left[\frac{1}{M+I}\left(\sum_{m=1}^{M}\frac{s_{m}^{y}}{y_{m}^{k^{\prime }}}+\sum_{i=1}^{I}\frac{s_{i}^{b}}{b_{i}^{k^{\prime }}}\right)\right]}\\ s.t.\left\{ \begin{array}{l} \sum_{k=1}^{K}z_{k}y_{m}^{k}-S_{m}^{y}=y_{m}^{k^{\prime }},m=1,\ldots ,M\\ \sum_{k=1}^{K}z_{k}b_{i}^{k}+S_{i}^{b}=b_{i}^{k^{\prime }},i=1,\ldots ,I\\ \sum_{k=1}^{K}z_{k}x_{n}^{k}+S_{n}^{x}=x_{n}^{k^{\prime }},n=1,\ldots ,N\\ z_{k}\geq 0,S_{m}^{y}\geq 0,S_{i}^{b}\geq 0,S_{n}^{x}\geq 0,k=1,\ldots ,K \end{array}\right. \end{array}$$

In the formula (3), $S^{x}$, $S^{y}$ and $S^{b}$ represent slack variables of input, output and pollution, respectively. The eco-efficiency, $\rho^{*}$, is the strictly decreasing objective function of the 3 slack variables, and the scale is [0, 1]. If $\rho^{*}$ = 1 and $S^{x}$ = 0, $S^{y}$ = 0, $S^{b}$ = 0, the DMU is fully efficient. If $\rho^{*}<1$ and $S^{x}$, $S^{y}$, $S^{b}$ are not all 0, the DMU is invalid, and the input or output should be improved.

Moreover, there is a problem that common DEA models, including the SBM model, cannot further rank the full-efficiency production units. Generally, multiple production units are needed to construct a valid frontier, so there are many full-efficient production units at the same time. This paper uses the SBM model to measure efficiency; on this basis, we no longer limit the efficiency of those full-efficiency production units to be equal to 1 and allow the efficiency to be greater than or equal to 1 so that the full-efficiency production units can be ranked. This kind of model is usually called the super-efficiency SBM model.

## 3. Variables and Data

### 3.1. Variable Description

This paper takes the provincial aquaculture industry as the subject of the research, brings the coordination of resources, the environment and development into the unified analytical framework, and conducts an empirical analysis with the SBM directional distance function model. According to the general analytical framework of aquaculture input and output, the selection of relevant variables and data statistics are described as follows.

1. Input Variables. Land, labor and fixed-assets are indispensable production factor inputs in agricultural production. According to previous studies, we chose aquaculturists, farming boats, farming area and aquaculture intermediate consumption as the input variables for aquaculture [24,25]. Among them, aquaculture intermediate consumption cannot be found directly in the statistical yearbook. Therefore, the intermediate consumption of the fishery is used for corresponding conversion. The calculation formula is as follows: Aquaculture intermediate consumption = intermediate consumption of fishery × (total output of aquaculture / total output of fishery), and taking the price index of 2003 as the base period, the calculation results are adjusted according to the price index of agricultural production so as to eliminate the impact of price changes.

2. Desirable Output. The desirable output is represented by the total output of aquaculture and is also adjusted according to the price index of agricultural production to eliminate the impact of price changes.

3. Undesirable Output. Due to the difficulty of statistics and estimation, the environmental yearbook usually does not provide relevant data for aquaculture pollution emissions. Thus, the accurate accounting of provincial aquaculture pollution emissions is the key point and difficulty of this paper. By comprehensively comparing various calculation methods and considering the availability of data, we finally chose the unit survey and assessment method to calculate the aquaculture pollution emission for each province. The unit survey and assessment method is a quantitative analysis method based on unit survey and unit analysis. In this study, we mainly refer to the calculation methods for agricultural pollution emissions used by Lai et al. (2004) and Li et al. (2011), and we adjust and modify the parameters, including pollution coefficient and emission coefficient of aquaculture [26,27].

Aquaculture pollution defined in this paper mainly refers to the production of COD_Cr_, TN and TP, which are generated in aquaculture, as well as the amount of these pollutants discharged into the surrounding waters through surface runoff, farm drainage and underground leaching. According to the type of aquaculture, the pollution source is divided into two units: mariculture and freshwater aquaculture, and the quantitative relationship between units, pollution production and pollution emission is established, as shown in formula (4):(4)$$E_{j}=\sum_{i}{EU}_{i}\rho_{ij}(1-\eta_{i})C_{ij}(EU_{ij},S)=\sum_{i}PE_{ij}\rho_{ij}(1-\eta_{i})C_{ij}(EU_{ij},S)$$

In formula (4), $E_{j}$ represents the emission of aquaculture pollutant j; ${EU}_{i}$ represents index statistics of unit i; $\rho_{ij}$ represents the coefficient of pollution production intensity of unit i; $\eta_{i}$ represents the coefficient of utilization efficiency of relevant resources; $PE_{ij}$ represents the production of pollutant j, termed the maximum pollution caused by aquaculture without considering factors such as comprehensive utilization and management of resources; and $C_{ij}$ represents the emission coefficient of pollutant j in unit i, which usually depends on the unit and spatial characteristic S and refers to the comprehensive impact of the environment, rainfall, hydrology and management measures of each province on aquaculture pollution. The production and emission coefficients of aquaculture pollution reference both the data summarized by the school of environment of Tsinghua University and also the “Manual of the National Pollution Source Survey on the Production and Emission Coefficients of Aquaculture Pollution Sources” and are calculated by comparing a large number of studies. As space is limited, the details are not be described here. With pollution production formula (4) and relevant data, the change in the total aquaculture pollution emissions for each province is obtained. In order to understand the emission of different pollutants as a whole, Figure 1 shows the emissions of major pollutants in China’s aquaculture industry from 2003 to 2018.The results show that the emission of different pollutants has increased year by year.

### 3.2. Descriptive Statistics

In the official statistical records such as the “China Fishery Statistics Yearbook” and “China Agriculture Yearbook”, the provincial output of China’s aquaculture has been calculated since 2003. Therefore, we chose the data from 2003 to 2018 as the subject of this study. In terms of regional selection, due to a serious lack of data in the four provinces of Tibet, Gansu, Qinghai and Ningxia, these provinces were excluded, and the other 27 regions are included in this research. The individual missing data in some provinces are filled in by the linear interpolation method. The descriptive statistics for all variables in this paper are shown in Table 1.

## 4. Results

According to the analytical framework, variable selection and data processing, this paper conducts an empirical analysis of the eco-efficiency of the aquaculture industry in 27 provinces of China from 2003 to 2018, comprehensively investigating the coordination among resources, the environment and aquaculture development in China.

### 4.1. Temporal Variations of Eco-Efficiency

Using the super-efficiency SBM model with and without undesirable outputs, respectively, the eco-efficiency and economic efficiency of China’s aquaculture industry from 2003 to 2018 are calculated. Through the comparison of the above results, the impact of the undesirable output of environmental pollution on the eco-efficiency of the aquaculture industry is studied, as shown in Figure 2.

The following findings can be obtained from Figure 2. First, the economic efficiency of China’s aquaculture is obviously higher than the eco-efficiency from 2006 to 2018. In addition, the eco-efficiency from 2003 to 2005 is slightly higher than the economic efficiency. Second, the eco-efficiency of aquaculture in China from 2003 to 2018 is 0.70, indicating the low level of eco-efficiency in China. Third, since 2012, the eco-efficiency of China’s aquaculture industry has trended up. Fourth, the eco-efficiency of China’s aquaculture industry from 2003 to 2005 is slightly higher than the economic efficiency.

### 4.2. Regional Changes of Eco-Efficiency

From regional distribution in Figure 3, the eco-efficiency of the aquaculture industry from 2003 to 2018 in eastern, central and western China is significantly different. The eco-efficiency of eastern China is the highest (0.98), followed by that of western China (0.67), and the eco-efficiency of central China is the lowest (0.55).

In terms of temporary changes, the eco-efficiency of aquaculture in eastern China from 2003 to 2018 was relatively stable and much higher than the average level eco-efficiency of China (0.70). The eco-efficiency of central China has been on a downward trajectory year by year. Especially since 2011, it has been at a low level with small fluctuations, and there is a remarkable gap between the eco-efficiency of central China and that of eastern or western China. The eco-efficiency of western China was activated at first and then inhibited, which is generally close to that of the whole country. Since 2012, the eco-efficiency of western China has been slightly higher that the average level of the whole country.

### 4.3. Ranking of Provincial Eco-Efficiency and Its Changes

In essence, the DEA model is used to evaluate the relative efficiency among DMUs, so this paper mainly analyzed the ranking of provincial eco-efficiency. When ranking eco-efficiency, the value of eco-efficiency is the arithmetic mean of the eco-efficiency of each province from 2003 to 2008. Table 2 shows the average of the eco-efficiency rankings for the provinces. The average eco-efficiency values for the six provinces or province-level municipalities of Fujian, Shanghai, Beijing, Hainan, Tianjin and Chongqing were greater than 1 from 2003 to 2018, indicating that the development of aquaculture in these areas has reached a relative coordination with resources and the environment. The average eco-efficiency values in the 11 provinces of Xinjiang, Guizhou, Jilin, Shandong, Guangdong, Sichuan, Guangxi, Inner Mongolia, Liaoning, Jiangxi and Yunnan were greater than 0.6 and less than or equal to 1, indicating that the development of aquaculture has been relatively uncoordinated with resources and the environment. The average eco-efficiency values for the 10 provinces of Zhejiang, Anhui, Hunan, Hebei, Jiangsu, Hubei, Henan, Heilongjiang, Shanxi and Shaanxi were less than 0.6, indicating that the development of aquaculture has been extremely uncoordinated with resources and the environment. Among the traditional provinces for aquaculture, only Fujian has been relatively coordinated.

Table 3 shows the dynamic changes in the eco-efficiency rankings. The table data indicates the ranking position in all provinces. It can be seen that the ranking for eco-efficiency in each province has basically changed little, and only a few provinces have undergone great changes. The eco-efficiency values for the 5 provinces of Fujian, Shanghai, Hainan, Tianjin and Chongqing have always been in the first echelon. The rankings for eco-efficiency in the 4 provinces of Xinjiang, Hubei, Zhejiang and Jiangsu have increased significantly, while the eco-efficiency values for Jilin and Inner Mongolia have decreased significantly. Additionally, the eco-efficiency rankings for Xinjiang have fluctuated greatly and decreased significantly during the 11th Five-Year Plan period but returned to the 4th ranking during the 12th Five-Year Plan period.

### 4.4. Production and the Best Practitioner

Generally, in the DEA analysis, when the efficiency of DMU is equal to 1, the production unit is on the production possibility boundary and is considered to be fully efficient. This paper can further evaluate and rank the full-efficiency production units with the super-efficiency SBM model. Therefore, the best practitioners of aquaculture eco-efficiency in this paper include not only the production units with efficiency of 1 but also those production units with efficiency greater than 1. See Table 4 for details.

From the dynamic changes in the eco-efficiency rankings, it can be seen that the ranking for eco-efficiency in each province has basically changed little, and only a few provinces have undergone great changes. The eco-efficiency values for the five provinces of Fujian, Shanghai, Hainan, Tianjin and Chongqing has always been in the first echelon. The rankings for eco-efficiency in the four provinces of Xinjiang, Hubei, Zhejiang and Jiangsu has increased significantly, while the eco-efficiency rankings for Jilin and Inner Mongolia have decreased significantly. Additionally, the eco-efficiency ranking for Xinjiang fluctuated greatly and decreased significantly during the 11th Five-Year Plan period but returned to the 4th rank during the 12th Five-Year Plan period.

## 5. Discussion

According to the above empirical analysis results, we can see that China’s aquaculture industry has the following characteristics.

First, on the whole, whether economic efficiency or eco-efficiency, the efficiency of China’s aquaculture industry is not high, which is consistent with the research of Zhang et al. (2014) [28]. Although China’s aquaculture industry has developed rapidly, there is much work to be done on resource conservation and environmental protection, and there is still a great incoordination among economic development, resources and the environment. On the other hand, the eco-efficiency of China’s aquaculture is obviously lower than the economic efficiency, indicating the undesirable outputs have a negative impact on aquaculture efficiency. The rapid development of aquaculture in China comes at the expense of the environment. This conclusion is basically consistent with the findings of Ji and Zeng (2016) and Qin et al. (2018). Ji and Zeng (2016) found that the eco-efficiency of mariculture in China was 0. 58 based on the data from 2003 to 2014. Qin et al. (2018) reported that the eco-efficiency of China’s mariculture from 2008 to 2016 turned out to be 0.62. The data structure of this paper is from 2003 to 2018. It is estimated that the eco-efficiency of aquaculture in China is 0.70, which shows that the results of this paper are generally reliable. The difference in results may be caused by the study object and data range.

Second, we found that the eco-efficiency had ascended step by step since 2012. After the Communist Party of China’s 18th National Congress, China incorporated ecological civilization into the “Five in One” overall layout, vigorously sped up the mode of transformation and structure adjustment of the fishery and continuously strengthened the management of the aquaculture water environment, thus promoting the eco-efficiency of the aquaculture industry. We also found that the eco-efficiency of China’s aquaculture industry from 2003 to 2005 was slightly higher than the economic efficiency. This phenomenon may be due to the fact that during the 10th Five-Year Plan period, provinces in China implemented aquaculture-based development policies, vigorously developed the cultivation of popular and high-quality aquatic products, and strictly controlled the development of fishing so that the eco-efficiency was relatively high in the early stages of the development of the aquaculture industry.

Third, the results of the empirical analysis show that the resources, the environment and the development of aquaculture in eastern China basically reach relative coordination. Western China shows a general coordination of aquaculture, although there is imbalance in aquaculture efficiency. Moreover, there is a serious imbalance in the aquaculture efficiency in central China, indicating that the development of aquaculture possibly comes at the cost of the environment and the breeding mode needs to be improved. With its resources and location advantages, the aquaculture industry in eastern China developed rapidly. At the same time, eastern China actively developed healthy aquaculture, continuously adjusted the aquaculture structure and vigorously improved the aquaculture infrastructure, facilitating high-quality development. The aquaculture industry in western China has made full use of its late development advantages under the background of promoting the high-quality development of aqua-culture, learning from the development experiences of advanced areas. The aquaculture industry in central China faces great pressure on environmental re-sources. Without timely changes in the development pattern of aquaculture, the resource and environmental constraints will reach their limits, leading to environmental problems.

## 6. Conclusions

This paper calculates the eco-efficiency to evaluate the coordination of environment and aquaculture growth based on the slacks-based measure directional distance function dealing with undesirable outputs, and analyzes the coordinated changes from 2003 to 2018. The conclusions are as follows.

First, the results of the empirical analysis show that the eco-efficiency in China’s aquaculture industry is not desirable. Environmental pollution has brought great efficiency loss to the development of aquaculture industry and failed to meet the requirement for the rapid and sound development of national economy. Eastern China shows the best coordination, followed by western China, and central China exhibits the most obvious imbalance among the environment, resources and aquaculture development.

Second, the dynamic changes in eco-efficiency reveal that 2003–2005 is the initial stage of vigorously developing aquaculture, and the eco-efficiency was relatively high during this period. However, since 2006, with the rapid development of aquaculture, the eco-efficiency showed a downward trend, and it did not show a upward trend until 2012. From the perspective of regional distribution, the eco-efficiency of eastern China has been relatively stable and has been maintained at a high level. The eco-efficiency of central China is at a low level all the year around and there has been no sign of significant improvement.

Third, the coordination of aquaculture development with the environment and resource among regions is extremely unbalanced. The relationship between aquaculture development and the environment in eastern China is relatively harmonious, with the six provinces or province-level municipalities of Fujian, Shanghai, Beijing, Hainan and Tianjin growing soundly and rapidly. However, the eco-efficiency in central and western China is generally low. The relationships between aquaculture and the environment in the five provinces of Hubei, Henan, Heilongjiang, Shanxi and Shaanxi are seriously unbalanced.

The above conclusions show that, while China’s aquaculture industry has made great achievements in development, it has faced great pressure on environmental resources. The overall coordinated degree of resource conservation, environmental protection and aquaculture development is not ideal with a certain imbalance, which is most obvious in central China. In addition, the eco-efficiency of most traditional aquaculture provinces is generally not satisfying. For the long-term growth of China’s aquaculture industry, on the one hand, it is necessary to transform technology into productivity and improve the technical level continuously. On the other hand, we should make efforts to improve management capacity and take full advantage of the potential of technology.

## Figures and Tables

**Figure 1 ijerph-19-08010-f001:**
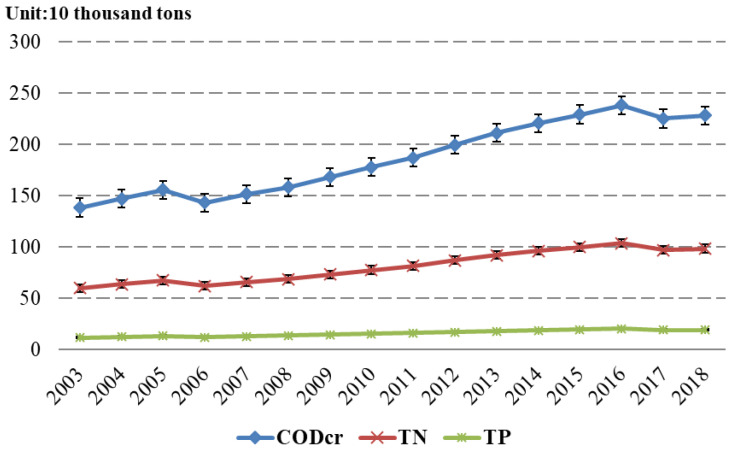
The emissions of major pollutants in China’s aquaculture industry from 2003 to 2018.

**Figure 2 ijerph-19-08010-f002:**
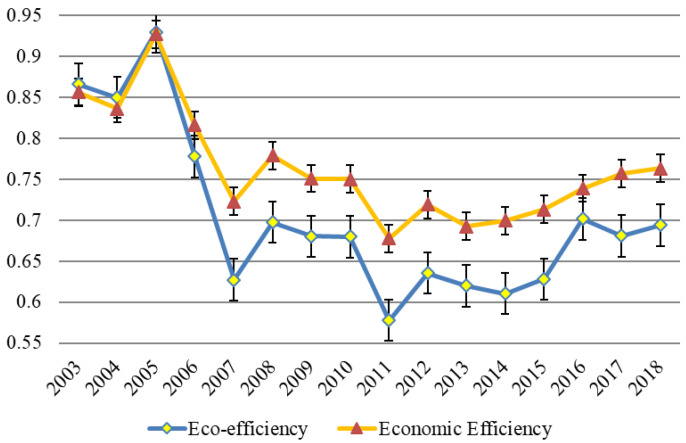
Eco-efficiency of aquaculture in China from 2003 to 2018.

**Figure 3 ijerph-19-08010-f003:**
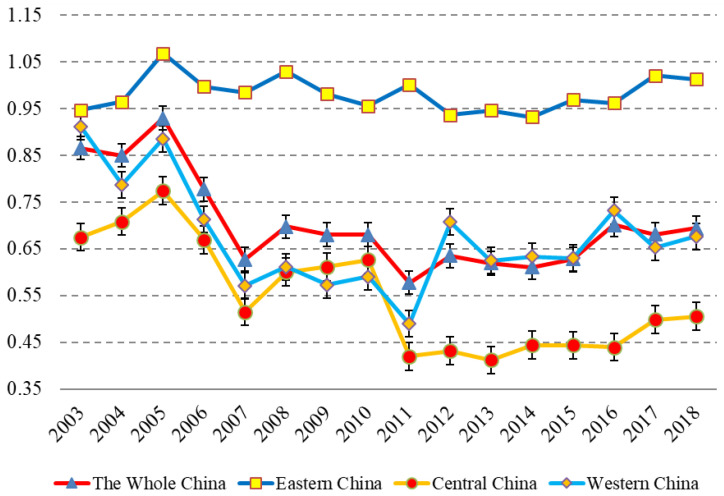
Eco-efficiency of aquaculture in eastern, central and western China from 2003 to 2018. (Notes: The eco-efficiency variable is the arithmetic mean of the eco-efficiency variable for each province. Eastern China includes 11 provinces or province-level municipalities: Beijing, Tianjin, Hebei, Liaoning, Shanghai, Jiangsu, Zhejiang, Fujian, Shandong, Guangdong and Hainan. Central China includes 8 provinces: Shanxi, Jilin, Heilongjiang, Anhui, Jiangxi, Henan, Hubei and Hunan. Western China includes 8 provinces or province-level municipalities: Inner Mongolia, Guangxi, Sichuan, Chongqing, Guizhou, Yunnan, Shaanxi and Xinjiang).

**Table 1 ijerph-19-08010-t001:** Descriptive Statistics.

Type	Variable	Minimum	Maximum	Mean	Standard Deviation
Input	Aquaculturist (people)	2014	772,921	180,813.70	184,040.65
Input	Farming Boats (kilowatt)	20	770,536	72,698.46	127,351.99
Input	Farming Area (hectare)	2606	1,152,153	270,216.39	252,360.69
Input	Aquaculture Intermediate Consumption (thousand yuan)	31,467.4	29,698,697.5	4,504,893.0	5,873,336.5
Desirable Output	Total Output of Aquaculture (thousand yuan)	531.6	69,026,842.2	11,721,380.4	14,564,475.3
Undesirable Output	Chemical Oxygen Demand (ton)	991.83	342,752.95	68,753.51	81,147.32
Undesirable Output	Total Nitrogen (ton)	680.29	127,521.05	29,946.01	33,096.08
Undesirable Output	Total Phosphorus (ton)	158.06	25,391.83	5885.34	6553.15
N = 27 × 16 = 432					

**Table 2 ijerph-19-08010-t002:** Average eco-efficiency for each province from 2003 to 2018.

Province	Average Eco-Efficiency	Ranking
Fujian	1.857	1
Shanghai	1.470	2
Beijing	1.407	3
Hainan	1.248	4
Chongqing	1.172	5
Tianjin	1.038	6
Xinjiang	0.812	7
Guizhou	0.763	8
Jilin	0.748	9
Shandong	0.734	10
Guangdong	0.704	11
Sichuan	0.686	12
Guangxi	0.671	13
Inner Mongoria	0.654	14
Liaoning	0.650	15
Jiangxi	0.644	16
Yunnan	0.634	17
Zhejiang	0.592	18
Anhui	0.565	19
Hunan	0.561	20
Hebei	0.552	21
Jiangsu	0.549	22
Hubei	0.537	23
Henan	0.477	24
Heilongjiang	0.434	25
Shanxi	0.423	26
Shaanxi	0.002	27

**Table 3 ijerph-19-08010-t003:** Dynamic changes in eco-efficiency rankings.

Province	2003–2005	2006–2010	2011–2015	2016–2018
Beijing	1	2	7	10
Fujian	2	1	1	1
Shanghai	3	3	2	2
Chongqing	4	6	5	3
Hainan	5	4	3	4
Jilin	6	5	25	25
Jiangxi	7	17	17	21
Yunnan	8	15	23	16
Guizhou	9	12	11	6
Guangxi	10	20	12	11
Sichuan	11	11	16	15
Inner Mongoria	12	9	21	19
Tianjin	13	7	6	5
Guangdong	14	14	9	7
Xinjiang	15	26	4	8
Anhui	16	21	19	17
Hunan	17	13	20	23
Heilongjiang	18	25	24	26
Henan	19	22	22	24
Zhejiang	20	16	13	13
Shandong	21	10	8	9
Hebei	22	18	18	18
Liaoning	23	8	10	20
Hubei	24	23	15	14
Shanxi	25	24	26	22
Jiangsu	26	19	14	12
Shaanxi	27	27	27	27

**Table 4 ijerph-19-08010-t004:** Best provincial practitioners of eco-efficiency from 2003 to 2018.

Period	Eastern China	Central China	Western China
2003–2005	Beijing(3), Fujian(3), Shanghai(3), Hainan(3), Tianjin(2), Guangdong(1)	Jilin(3), Jiangxi(3)	Chongqing(3), Yunnan(3), Guizhou(3), Guangxi(3), Sichuan(2), Inner Mongoria(2), Xinjiang(1)
2006–2010	Fujian(5), Beijing(5), Hainan(5), Shanghai(5), Tianjin(5), Liaoning(3), Shandong(2)	Jilin(5), Jiangxi(1), Hunan(1)	Chongqing(5), Guizhou(2), Inner Mongoria(2), Sichuan(1), Yunnan(1), Xinjiang(1)
2011–2015	Fujian(5), Shanghai(5), Hainan(5), Tianjin(5), Beijing(4), Shandong(1)	—	Chongqing(5), Xinjiang(4), Guizhou(1)
2016–2018	Fujian(3), Shanghai(3), Hainan(3), Tianjin(3), Shandong(1), Beijing(1)	—	Chongqing(3), Guizhou(2), Guangxi(1), Xinjiang(1)

Notes: When ranking eco-efficiency, the value for eco-efficiency is the arithmetic mean of the eco-efficiency of each province in the corresponding period. The data in brackets indicates the change of ranking relative to the previous ranking. Specifically, a positive number indicates that there was a rise, a negative number indicates that there was a decline, and zero indicates there was no change.

## Data Availability

Not applicable.

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
