# Peer review of "The Coordination of Aquaculture Development with Environment and Resources: Based on Measurement of Provincial Eco-Efficiency in China"

_ijerph, 2022, doi:10.3390/ijerph19138010_

Round 1

Reviewer 1 Report

The manuscript by Wen and Changbiao is interesting and meets the scientific quality to this journal. The research is well designed and the result supports the conclusion. Overall, the manuscript has merit to be published in this journal. I have some minor concerns, which are as follows

Abstract missing neumerical value

Some references are too old. I would prefer not to use any ref. more than 20 years old.

Figure 1, 2, 3: Missing error bar

Check the ref (citation and bibliography).

Reviewer 2 Report

The article needs major revisions. Detailed directions and recommendations are below:

1. The TITLE needs to be made more specific. It should better reflect the objectives of the paper and the area of study.

2. The ABSTRACT needs to be rewritten and completed. It is written what was done in the paper and what methods were used. However, it should be explicitly written what was the aim of the research.

3. The INTRODUCTION is currently the best chapter of the article. However, it should end with a paragraph with precisely stated research objectives. This should be completed in detail. It is not clear from the description whether the research was on the entire aquaculture industry in China (freshwater + marine)? The text in lines 98-115 is more appropriate for the METHODS chapter.

4. METHODS - make a small correction to the text. Please clearly state what methods (models) were used to achieve a specific goal and justify why.

5. VARIABLES AND DATA - The data shown in Fig. 1 are very important to the final results. It appears from the text that they were calculated using a formula. Thus, in my opinion, they are therefore part of the RESULTS. The description of the calculation of these data needs to be more detailed. Fig 1 shows data for the whole aquaculture industry. However, the text (line 209-210) writes that the calculations are for each province. This needs clarification. Overall, there is a lack of insight into the detailed data.

6. RESULTS AND DISCUSSION - this section needs the most revision. In fact, only the results presented in two Tables and two Figures are included. No discussion has been made. No confrontation with the results of other authors' studies has been made. Only line 312 says that the results are consistent with Qin Hong et al. (2018). This is too little. In general, I recommend preparing the DISCUSSION as a separate chapter. Table 2 shows the ranking, but in a very unreadable way. This Table needs to be cleaned up. All the "notes" under the figures and tables should be explained in the METHODS chapter.

7. CONCLUSION - This is currently a summary of the results. This chapter should be completed. It should be based on the conclusions of a well-conducted discussion.

Round 2

Reviewer 2 Report

A large number of the suggested revisions were made. In the last paragraph of the INTRODUCTION section, the objectives of the study were described. Minor factual errors in the text of the paper have been corrected. The data in the tables have been organized. However, the correction of the TITLE is unsatisfactory. Indicating that the paper is about the ranking of indicators in the provinces is not enough. Can't it be written that the study is about China? I note that this is not even indicated by the KEYWORDS. Please think more deeply about this problem. Unfortunately, the authors did not consider the need for a deep correction and rewriting of the RESULTS AND DISCUSSION section. There is no scientific discussion in the article. Therefore, I recommended the creation of a separate DISCUSSION chapter. I note that in the INTRODUCTION chapter there are references to literature items that are perfectly suited to induce such a discussion. The obtained results should be confronted with other publications that refer to phenomena of a similar nature in other countries or regions of the world. At present, the article is only a report on changes in the eco-efficiency index in individual provinces and ranks them. This is too little for the status of the journal.

In response to my comments, the authors only wrote that "Figure 2 shows the confrontation with the results of other authors' studies has been made". This cannot be agreed with. Moreover, there are no references to the literature. A summary of the temporal changes in the coefficient values is part of RESULTS, but not a scientific discussion. Once again, I recommend having a good scientific discussion and forming solid conclusions based on it. This will significantly improve the quality of this article.
